# Dust Exposure and Respiratory Health among Selected Factories in Ethiopia: Existing Evidence, Current Gaps and Future Directions

**Akeza Awealom Asgedom**

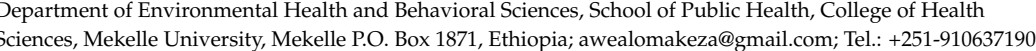

Department of Environmental Health and Behavioral Sciences, School of Public Health, College of Health Sciences, Mekelle University, Mekelle P.O. Box 1871, Ethiopia; awealomakeza@gmail.com; Tel.: +251-910637190

**Abstract:** Workers who are working in dusty environments might be associated with respiratory health problems. In Ethiopia, factories processing wood, textile, coffee, flour, cement and other materials are associated with dust emission. Furthermore, despite the adoption of the International Labor Organization (ILO) convention, the available constitution and labor proclamation, there are a lot of gaps in terms of occupational health and safety measures in Ethiopia. The current review aims to examine the existing evidence, current challenges and future direction regarding dust exposure and respiratory health in selected Ethiopian factories. Searches of peer-reviewed articles with full-length papers were made in online databases such as PubMed, Web of Science, MEDLINE, EMBASE and Google Scholar with a key words "Dust exposure", "Respiratory health", "Respiratory symptom", "Ethiopia" and "Factory workers" from January 2000 to March 2023. The review found that excessive dust exposure is associated with a high prevalence of respiratory health problems. Lack of personal protective equipment and absence of safety and health training were the main occupational health deficits identified in most factories. Actions that focus on these deficiencies are commendable. Interventions focused on safety and health trainings, and the provision of adequate personal protective equipment of the required quality is recommended. In addition, administrative solutions and longitudinal studies on dust exposure and respiratory health are suggested.

**Keywords:** dust; exposure; Ethiopia; factory; health; respiratory





## 1. Introduction

Globally, around 12.6 million people die as a result of living or working in an unhealthy environment, accounting for 25% of all total global deaths [1]. Outdoor air pollution caused 4.2 million premature deaths worldwide per year in 2019 as a result of exposure to fine particulate matter [2].

Occupational exposure to dust is significantly associated with respiratory health problems [3]. Duration of exposure usually determines the likelihood of the outcome. Dust exposure even below the recommended exposure limit results in respiratory health complaints [4,5].

Respiratory diseases can cause suffering for workers and their families and can be costly for the employer as a result of increased absenteeism, reducing productivity of the factory and associated medical costs for the sick workers. Occupational exposure is a potential cause of almost all respiratory diseases, but the majority of occupational-related respiratory health problems are often under-reported, and recognition of occupation-related respiratory-health problems can be increased by expanding their epidemiology through comprehensive research [6].

The economic cost of work-related injury and illness is 1.8–6% of Gross Domestic Product (GDP) in country estimates, with an average of 4% of GDP [7]. Hence, safety and health at work remain a major global concern. As a result, urgent action is required to reverse this trend of workplace accidents and diseases [8].

The International Labor Organization's (ILO) statement indicates that insufficient awareness and understanding of occupational hazards and risks, a lack of the requisite capacity for prevention, compliance and enforcement related to Occupational Safety and Health (OSH), inadequate and inaccurate data on occupational fatalities, injuries and diseases and inadequate legislation, regulations and policies on OSH remain significant challenges, particularly in developing countries, including Ethiopia [9]. However, these occupation-related health problems can be prevented by applying engineering solutions and administrative measures and through use of personal protective equipment.

Despite the various occupational health problems, Ethiopia has been a member state of the International Labor Organization (ILO) since 1923 and has ratified 20 conventions. The Occupational Safety and Health Convention, 1981 (No. 155), is one of the conventions ratified for the protection of workers in their work environment [10]. The government of Ethiopia included promotion of occupational health and safety as a priority issue in its health-policy statement released in 1993 [11]. The government of Ethiopia has also issued a labor proclamation (Proclamation No. 377/2003), which addresses the occupational safety and health measures in Section 3, 'Obligations of an Employer' Article 12/4 [12]. The issue of occupational safety and health is also included in the constitution of the country, with the right to a healthy and safe work environment being stated in Article 42/2 [13]. However, there is no national occupational safety and health policy that promotes the health and safety of workers in all occupations [10].

Despite the adoption of the ILO convention, the available Ethiopian constitution and the labor proclamation, there are a lot of gaps in terms of occupational health and safety measures in Ethiopia [14]. The performance of the manufacturing sector and other factories is affected because it uses outdated technologies, poor state of the work environment, limited research and development, poor institutional frame-work and inadequate managerial and technical skills [15].

According to Global Burden of Disease, occupational and environmental factors are the top risk factors that drive death and disability in Ethiopia [16]. Occupational factors had a 9.24% contribution rate for lung cancer, 100% for pneumoconiosis, 17.56% for chronic obstructive pulmonary disease (COPD) and 9.34% for asthma. However, when other environmental factors were combined with the occupational risk factors, it had a 49.22% contribution rate for lung cancer and 51.91% for chronic obstructive pulmonary disease (COPD) [17].

There is a paucity of evidence regarding occupational safety and the health situation in Ethiopia despite the few published studies. The current review examines the existing evidence, current challenges and future direction about dust exposure and respiratory health in selected Ethiopian factories. This will help for evidence-based intervention, policy consideration and possible research direction in the future.

## 2. Materials and Methods

### 2.1. Search Strategy

A search of articles was made in online databases such as PubMed, Web of Science, MEDLINE, EMBASE and Google Scholar with a key words "Dust exposure", "Respiratory health", "Respiratory symptom", "Ethiopia" and "Factory workers" from January 2000 to March 2023. This is to include the relevant but limited studies conducted in this period.

### 2.2. Inclusion and Exclusion Criteria

Full length peer-reviewed articles showing either dust exposure and respiratory health consequences or respiratory health in dusty work environments were considered for this review. In addition, articles that show either dust exposure or respiratory health but are by the same authors and discuss dust exposure and respiratory health separately were considered for review. However, articles that only reflect exposure studies without evaluating respiratory health issues were excluded from the review. In addition, articles that address respiratory health problems of factory workers in a dusty work environment

without evaluating exposure were considered for this review. The reviewed article includes factories that generate dust due to the nature of the work process, such as wood, coffee, cement, textile, flower, flour, tannery, agriculture, paper, and steel. These workplaces were included because of their available relevant data for the objective of this review.

*2.3. Search Process Flow and Results*

A total of 27 papers from 10 different factories were considered for this review. There were 4 articles from the wood industry, 5 from the textile industry, 4 from the Coffee industry, 4 from the cement industry, 3 from the flour industry, 2 from the flower industry, 2 from agriculture and 1 article each from the paper, the tannery and the steel industries found to be used for this review.

### 3. Results

*General Information of the Reviewed Papers*

As shown in Table 1, few studies have been conducted on dust exposure and respiratory health in different Ethiopian work environments. The main findings of these studies show that high exposure to dust from wood, coffee, cement, textile, flour, flower, paper, tannery, agriculture and steel is associated with a high prevalence of respiratory health problems. In addition, workers in most work environments do not use appropriate and adequate safety materials. There is also no health and safety training for the workers in most of the occupations studied. This affects worker health and safety by increasing absenteeism and medical costs and decreasing factory productivity.

**Table 1.** Summary of main findings on dust exposure and respiratory health done in selected Ethiopian factories.

| Factory | Study Design | Sample Size | Variable of Study | Main Findings | Scale | Reference |
|---------|--------------|-------------|-------------------|---------------|-------|-----------|
| Wood | Cross sectional | 147 | Respiratory symptom | • High prevalence of respiratory symptoms (24–45%) as compared to the control groups (2.7–15%). • No significant difference on lung function status. • No proper personal protective equipment (PPE). • No safety and health training. | Large scale | [18] |
| | | 152/45 | Exposure | • GM Inhalable dust exposure ($n = 152$) = 4.66 mg/m$^3$. • GM Endotoxin exposure ($n = 152$) = 62.2 EU/m$^3$. • Formaldehyde exposure ($n = 45$); ranges < 0.2–5 ppm. • No proper personal protective equipment (PPE). • No safety and health training. | Large scale | [19] |

**Table 1.** *Cont.*

| Factory | Study Design | Sample Size | Variable of Study | Main Findings | Scale | Reference |
|---|---|---|---|---|---|---|
| Wood | Cross sectional | 172 | Knowledge, Attitude and Practice (KAP) | • Permanent workers had better knowledge and attitude towards chemical hazards and PPE.<br>• PPE were not as per the required quality.<br>• There was poor PPE practice.<br>• No safety and health training. | Large scale | [20] |
| | | 506 | Exposure and Respiratory symptoms | • GM Inhalable dust exposure = 10.27 mg/m$^3$.<br>• High prevalence of respiratory symptoms (42.1–54.6%).<br>• Work experience, using bio-fuel as an energy source for cooking, past occupational dust exposure history and having no safety and health training were identified risk factors. | Medium scale | [21] |
| Coffee | Cross sectional | 360 | Exposure | • GM Total personal dust exposure varies from 1.08 to 12.54 mg/m$^3$. | Large scale | [22] |
| | | 225 | Respiratory Health | • High prevalence of respiratory symptoms (17.9–46.4%) as compared to the controls (1.9–11.3%).<br>• Reduced lung function among exposed workers as compared to the controls (water bottling workers).<br>• No safety and health training. | Large scale | [23] |
| | | 36 | Microbial contamination | • A microbial load ranged from $2.5 \times 10^2$ to $7.2 \times 10^5$ colony forming units (CFU)/mL.<br>• Presence of Gram-negative bacteria.<br>• No safety and health training. | Large scale | [24] |
| | | 549 | Respiratory Health | • Hand picker experienced a higher prevalence of respiratory symptoms as compared to controls.<br>• No personal protective device. | Large scale | [25] |

**Table 1.** *Cont.*

| Factory | Study Design | Sample Size | Variable of Study | Main Findings | Scale | Reference |
|---|---|---|---|---|---|---|
| Flower farm | Cross sectional | 273 | Respiratory symptoms | • Limited access to personal protective equipment (PPE). <br> • Unsafe pesticide routines. <br> • High prevalence of respiratory symptoms. <br> • No safety and health training. | Large scale | [26] |
| | | 248 | Respiratory symptoms | • Respiratory symptoms vary from 4–53%. <br> • No safety and health training. | Large scale | [27] |
| Cement | Cross sectional and follow up | 60 | Exposure and Respiratory Health * | • GM of Total dust varies from 0.4 mg/m$^3$ to 38.6 mg/m$^3$. <br> • Prevalence of respiratory symptom varies from 45–85%. | Large scale | [28] |
| | | 262 | Respiratory Health ** | • GM of Total dust exposure varies from 8.2 mg/m$^3$ to 432 mg/m$^3$. <br> • Respiratory symptoms vary from 2.6–73.7%. <br> • Reduction in lung function. | Large scale | [29] |
| | | 150 | Exposure * | • GM Total dust varies 153–549 mg/m$^3$. <br> • No safety and health training. | Large scale | [30] |
| | | 404 | Respiratory Health * | • Prevalence of respiratory symptoms ranges from 21% to 38.6%. <br> • No proper personal protective equipment (PPE). <br> • No safety and health training. <br> • Sex, age, educational status, work section, work experience, safety and health training, smoking and chronic respiratory diseases were the identified determinant factors. | Large scale | [31] |
| Textile | Cross sectional | 550 | Respiratory Health | • The prevalence of self-reported respiratory symptoms was 47.8% as compared to controls 15.3%. <br> • No safety and health training. <br> • Sex, service year, ventilation and working section were the identified determinant factors. | Large scale | [32] |
| | | 96 | Exposure | • GM Inhalable dust (GM)′ = 0.75 mg/m$^3$. <br> • GM Endotoxin (GM) = 831 EU/m$^3$. <br> • No proper personal protective equipment (PPE). <br> • No safety and health training. | Large scale | [33] |

**Table 1.** *Cont.*

| Factory | Study Design | Sample Size | Variable of Study | Main Findings | Scale | Reference |
|---|---|---|---|---|---|---|
| Textile | Cross sectional | 462 | Respiratory Health | • The prevalence of respiratory symptoms was 23–37% as compared to controls (5–17%).<br>• Higher cross-shift lung function reduction was observed among textile workers (123 mL for FEV1 and 129 mL for FVC) as compared with the control group (14 mL for FEV1 and 12 mL for FVC). | Large scale | [34] |
| | | 384 | Respiratory health | • The prevalence of respiratory symptoms varies from 0.5% to 11.7%.<br>• Educational level, working section and PPE utilization were the determinant factors. | Large scale | [35] |
| | | 7,992 | Registered health problems | • Respiratory diseases (34%) and musculoskeletal disorders (29%) were the most prevalent diagnoses.<br>• 16,993 absent working days due to sick leave were reported.<br>• Work department, sex and educational status were the determinant factors. | Large scale | [36] |
| Tannery | Cross sectional | 602 | Respiratory Health | • The prevalence of respiratory symptoms among exposed workers was 27.1% while it was 8.3% among unexposed workers.<br>• The odds of developing respiratory symptoms were 3.37 times higher among tannery workers than unexposed workers (civil servants).<br>• Sex, employment status, ventilation, safety and health training and PPE utilization were the determinant factors. | Large scale | [37] |
| Agriculture | Cross sectional | 1104 | Respiratory Health | • Increased risks for respiratory symptom (OR = 3.15 to 6.67) among the exposed subjects as compared with unexposed.<br>• Reductions in $FEV_1$ (140 mL), forced expiratory flow 25–75% (550 mL/s) and risk of $FEV_1$/FVC ratio < 0.8.<br>• No safety and health training. | Large scale | [38] |
| | | 334 | Respiratory health | • Reduction of pulmonary function and frequent complaints of respiratory symptoms among farm workers. | Large scale | [39] |

**Table 1.** *Cont.*

| Factory | Study Design | Sample Size | Variable of Study | Main Findings | Scale | Reference |
|---|---|---|---|---|---|---|
| Flour | Cross sectional | 406 | Respiratory health | • Prevalence of chronic respiratory symptoms was higher among flour mill workers (11.9–56.6%) as compared to soft-drinks factory workers (2.9–12.9%). • Reduction of lung function among flour mill workers as compared to controls. • Educational level, working section, work experience and working over eight hours were the determinant factors. | Large scale | [31] |
| | | 108 | Respiratory health | • Reduction in pulmonary function in flour mill workers as compared to their matched controls. • Prevalence of chronic respiratory symptoms was higher among flour mill workers (11.1–27.7%) as compared to controls (3.8–9.3%). | Large scale | [40] |
| | | 424 | Respiratory health | • Prevalence of chronic respiratory symptoms varies from 14.2 to 58.3%. • Age, income, work experience, past dust exposure and PPE utilization were the determinant factors. | Large scale | [41] |
| Paper | Cross sectional | 40 | Exposure | • GM of dust exposure = 10.2 mg/m$^3$ | Large scale | [42] |
| | | 434 | Respiratory health | • Prevalence of chronic respiratory symptoms varies (17–32.5%) as compared to controls (4.4–8.9%). • Educational status, working section, work experience and working more than 8 h per day were the determinant factors. | Large scale | [42] |
| Steel | Cross sectional | 75 | | • Mean particulate matter level = 153.7–2927 μg/m$^3$. • Respiratory symptom varies from 25–39%. | Large scale | [43] |

EU/m$^3$: endotoxin unit per cubic meter; GM: Geometric Mean; *n*: sample size; mg/m$^3$: milligram per cubic meter; μg/m$^3$: microgram per cubic meter; OR: Odds Ratio; PPE: personal protective equipment; KAP: Knowledge, Attitude and Practice; *: Cross sectional study; **: Follow up study; FVC: Forced Vital Capacity; FEV1: Forced Expiratory Volume in one second.

## 4. Discussion

### 4.1. Study Design

The study design of the reviewed manuscripts is mostly cross sectional, with the exception of a follow up study conducted in a cement factory [29]. Both dust exposure and respiratory health assessments were conducted in wood, coffee, textile, cement, paper and steel factories. However, only respiratory health has been studied in flower, flour, tannery and agricultural work environments. Respiratory health could be a proxy indicator of the possibility of high dust exposure in these factories.

*4.2. Existing Evidence*

The type of dust emitted varies from factory to factory depending on the nature of the process and the raw material used. For example, the wood industry produces wood dust, coffee industry coffee dust, textile industry cotton dust, cement industry silica dust, flour industry flour dust, tannery industry buffing dust, steel industry metallic dust, etc.

The overall results of this review indicate that dust exposure above the recommended limit is associated with a high prevalence of respiratory health problems, which is consistent with other findings [3]. However, the exposure limit standard varies from factory to factory depending on the type of dust emitted, and the studies adopt exposure standards from different countries.

For example, elevated levels of total dust and inhalable dust have been measured in medium- and large-scale wood factories [19,21]. In addition, high endotoxin and formaldehyde levels have been found in the large-scale wood industry [19]. Dust, endotoxin and formaldehyde exposure of workers in such factories has been associated with a high prevalence of respiratory health problems. Personal protective equipment, safety and health training were not provided [18,21]. This finding is consistent with studies conducted in the small-scale wood industry outside Ethiopia [44].

Excessive dust exposure and microbial contamination with Gram-negative bacteria was found in large-scale coffee processing factories [22,24], which was associated with a high prevalence of respiratory health problems [23,25]. This finding is consistent with the result of another study conducted outside Ethiopia [45,46].

High dust levels have been measured in the cement factory, which are associated with respiratory health problems [28,30]. The results are consistent with other studies conducted elsewhere [47,48]. The results are conclusive for the longitudinal studies showing decreased lung function in a follow-up study [29].

In textile factories, higher dust exposure was associated with higher prevalence of respiratory health problems [32–36]. The results are consistent with other studies [49].

In paper and steel factories, excessive dust exposure with a high prevalence of respiratory health problems was reported [42,43]. The results are similar with those of other studies [50,51].

A higher prevalence of respiratory health problems has been reported in flour, flower, tannery and agricultural work environments [26,27,31,40,41]. However, exposure studies have not been conducted in these workplaces. However, due to the nature of the work process, dust emissions are expected in such workplaces, which could lead to a high prevalence of reported respiratory problems among the workers. There is supporting evidence that has been conducted for flour and tannery industries [52,53]. Longitudinal exposure studies could be considered as an area of research for interested investigators to examine the relationship between dust exposure and respiratory health in such workplaces.

The most common predictors of respiratory health problems identified in most workplaces were socio-demographic factors such as gender, age, educational status and income. Behavioral and workplace factors such as work area, work experience, hours worked in excess of 8 h per day, employment status, previous dust exposure, ventilation, use of biofuel as an energy source, smoking, chronic respiratory disease, use of PPE and safety and health training must be considered to determine if interventions are needed.

In addition, Ethiopia has weak internal capacity to monitor and assess workplace hazards [14]. There is also a lack of standards and regulations for dust exposure and control mechanisms in various factories, which could exacerbate the occupational safety and health problem. Overall, the current review shows that inadequate implementation of occupational safety and health leads to excessive dust exposure and a higher prevalence of respiratory problems.

*4.3. Current Challenges*

In most of the articles reviewed, attention to occupational safety and health is considered one of the major challenges of most workplaces. This is a neglected problem that

requires due attention. There is no or limited supply of personal protective equipment and safety and health training for workers in various occupations.

### 4.4. Future Perspectives

The results of the present review are based on the design of a cross sectional study, which has the limitations of cross-sectional studies. The results are inconclusive regarding the effects of dust exposure on respiratory health. Although the results are based on a cross-sectional design, the level of exposure, the high prevalence of respiratory illness associated with the lack or limited supply of personal protective equipment and the lack of occupational safety and health training indicate the need for further intervention.

To improve worker safety and health, a hierarchy of workplace hazard control from most effective to least effective can be considered, depending on the factory's circumstances in terms of resources and feasibility. The hierarchy can be described as follows: Elimination, Substitution, Engineering Control, Administrative Control and PPE [54]. To reduce dust exposure and ensure worker respiratory health, the most effective control measures may not be in place or may not function adequately. Therefore, in many workplaces, administrative measures and PPE are recommended as immediate control measures. However, this is considered a last resort to protect against occupational hazards in the workplace.

## 5. Conclusions

This review found that excessive dust exposure is associated with a high prevalence of respiratory health problems. Lack of personal protective equipment and lack of safety and health training were the main occupational health deficits identified in most factories (more than thirteen articles). Actions that focus on these deficiencies are commendable. Interventions focused on safety and health training and the provision of adequate personal protective equipment of the required quality are recommended. In addition, administrative solutions and longitudinal studies on dust exposure and respiratory health are suggested.

**Funding:** This research received no external funding.

**Institutional Review Board Statement:** Not applicable.

**Informed Consent Statement:** Not applicable.

**Data Availability Statement:** Not applicable.

**Conflicts of Interest:** The author declares no conflict of interest.

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
