# Peer review of "Dust Exposure and Respiratory Health among Selected Factories in Ethiopia: Existing Evidence, Current Gaps and Future Directions"

_2673-527X, doi:10.3390/jor3020006_

Round 1

Reviewer 1 Report

Reviewer comments:

The redaction structure in the introduction is very good, however is important mention the international context that occur in other countries about the dust exposure and respiratory health, not only the Ethiopia situation. Is important introduce to reader the international context of the dust generated by factories and the health problemas.

For the Materials and Methods section is important to justify why the study period is from January 2000 to March 2023.

For the Materials and Methods section is important to justify why the study period is from January 2000 to March 2023. Aditionally, is important explain why was selected only certain types of factories and not others, if this is related to the economic activities realized in Ethiopie, is important explaine it in this section.

In Table 1 its no clear the meaning of heading "Remark", due to in the column its describe the size of some type of scale. Also its no clear if the scale in that column is spatial or temporal.

It is recommended describe the headings "Study design" and "Remark". Also, please describe in which consist the "Cross sectional" and "follow up" methods.

In the "Discussion" section is important to deep in the regulation for dust emissions or/and dust concentrations, and their relation with healthy problems.

Lines 126-131: It is recommended mention other air quality studies in which had been occupied the methods of design.

Lines 133-165: Please compare the evidences finded for Ethiopia with other studies that had showed similar conditions. The comparisson with other studies will help to rich the redaction too in the sections 4.3, 4.4 and Conclusions.

Author Response

Dear Reviewer,
First, we would like to acknowledge for the indispensable comments given that improves the quality of our manuscript. We will address the points step by step. The changes made are in track changes on the revised review manuscript.

Reviewer 2 Report

This manuscript presents a review of published literature on occupational dust exposure and respiratory disease in Ethiopia. Studies were summarized by type of industry, study design and relevant findings.  The authors also  identify specific gaps in hazard awareness and training,  lack of exposure controls and inconsistent focus on workplace safety and health programs.

Overall, the manuscript is well written but could use a careful review for English.  There are some minor  issues with grammar.

A careful update of web links is required and if possible, please provide more information of the actual document citation  and publication date. Page numbers of specific reference is useful. 

A marked-up copy of the draft is attached with specific comments

Author Response

(The authors gave the same response as above.)

Reviewer 3 Report

Review

General Comments

The manuscript entitled “Dust exposure and respiratory health among selected factories in Ethiopia: Existing evidence, current gaps and future directions” conduits a review on research regarding the occupational safety and health within factories known for dust exposure problems within Ehiopia. Nevertheless, I believe this review manuscript had to overcame challenging conditions since the 27 papers in which is based are very different between each other and I do not believe it reach entirely the author’s initial intention. First, this review paper has a very solid and contextual introduction about the occupational safety and health on Ethiopia. The Materials and Methods sections describes the wide scope of the criterium for choosing related research for the review. Since almost every study selected follows a cross sectional design, is hard to find proper relationships between variables, and Discussion section deals with this. For example, while studies mention the relation between dust exposure, respiratory problems and the paper factories, this is not completely clear in the case of fluor, since only 9 of the 27 selected studies are focused in dust concentrations. This forces the review paper to indirectly highlight the relationship between certain industries and respiratory problems, instead of dust exposure and respiratory problems. While the discussion section suggests using the respiratory health as a proxy for some industries dust conditions, I believe it could be made a little more. For example, referencing studies made in other similar countries focused in the same industries, which could have information about dust conditions that could be assumed as similar to the factories present in Ethiopia. I believe this review paper is very important to highlight the lack of research in this area but it can only be considered to be published after a major revision. I believe the review paper should be focused in resolving critical questions to understand its purpose. For example, Is there a relation between dust exposure and respiratory health problems in certain factories of Ethiopia? Is there a way to estimate the industry with the most problems? Which is the common point between the possible problems within these factories (e.g., lack of PPE)? Finally, there is some rewording needed in several parts of the manuscript and the english could be improved.

Specific comments:

Line 12 Specify ILO meaning from here.

Line 17 “with the key words”

Line 26 Keywords are repetitive and too many.

Table 1 Specify geometric mean (GM) acronym.

Line 177 – 181 This part is redundant.

Line 183 I believe this review shows an association between high prevalence of respiratory health problems and certain kind of industries. This could be overcome using reference from studies associating these industries (wood, coffee, textile, etc…) with high dust presence, even studies from other countries.

Line 184 How many studies highlighted this?

Line 186 – 188 This should be reworded.

Author Response

(The authors gave the same response as above.)

Round 2

Reviewer 3 Report

The author carried out every single suggestion from the previous revision round. Is evident that the manuscript has been under major revisions  thoroughly and has been greatly improved by that. Many references were added giving to the manuscript a new global context. I recommend the publication in this current state.